



# Estimation of fire-induced carbon emission from Equatorial Asia in 2015 by using in situ aircraft and ship observations

Yosuke Niwa[1,2], Yousuke Sawa[2*], Hideki Nara[1], Toshinobu Machida[1], Hidekazu Matsueda[2**], Taku Umezawa[1], Akihiko Ito[1], Shin-Ichiro Nakaoka[1], Hiroshi Tanimoto[1], and Yasunori Tohjima[1]

[1] National Institute for Environmental Studies, Tsukuba, Japan
  [2] Meteorological Research Institute, Tsukuba, Japan
  *Now at Japan Meteorological Agency, Tokyo, Japan
  ** Now at Dokkyo University, Soka, Japan

*Correspondence to* Yosuke Niwa (niwa.yosuke@nies.go.jp)

**Abstract.** The inverse analysis was used to estimate the fire carbon emission in Equatorial Asia induced by the big El Niño in 2015. This inverse analysis is unique because it extensively used high-precision atmospheric mole fraction data of carbon dioxide ($CO_2$) from the commercial aircraft observation project. By comparisons with independent shipboard observations, especially carbon monoxide (CO) data, the validity of the estimated fire-induced carbon emission was elucidated. The best

estimate, which used both aircraft and shipboard $CO_2$ observations, indicated 273 Tg C for fire emission during September–October 2015. This two-month-long emission accounts for 75% of the annual total fire emission and 45% of the annual total net carbon flux within the region, indicating that fire emission is a dominant driving force of interannual variations of carbon fluxes in Equatorial Asia. Several sensitivity experiments demonstrated that aircraft observations could measure fire signals, though they showed a certain degree of sensitivity to prior fire-emission data. The inversions coherently estimated smaller fire

emissions than the priors, partially because of the small contribution of peatland fires, indicated by enhancement ratios of CO and $CO_2$ observed by the ship. In the future warmer climate condition, Equatorial Asia would experience more severe droughts and have risks for releasing a large amount of carbon into the atmosphere. Therefore, the continuation of aircraft and shipboard observations is fruitful for reliable monitoring of carbon fluxes in Equatorial Asia.

## 1 Introduction

Equatorial Asia, which includes Indonesia, Malaysia, Papua New Guinea and surrounding areas (Fig. 1) has experienced extensive biomass burnings, especially during drought conditions induced by El Niño and the Indian Ocean dipole (Field et al., 2009). Those biomass burnings have emitted a significant amount of carbon, mainly in the form of carbon dioxide ($CO_2$), into the atmosphere (Page et al., 2002; Patra et al., 2005; van der Werf et al., 2008). Furthermore, those fire-induced carbon emissions in Equatorial Asia came from peatland, which has a remarkably high carbon density. Since the peatland in Equatorial

Asia accounts for a significant portion of the global peatland (Page et al., 2011), the region has a distinct role in the global carbon cycle despite its small terrestrial coverage.



In 2015, the extreme El Niño, accompanied by a positive anomaly of the Indian Ocean dipole mode, induced severe drought and devastating biomass burnings in Equatorial Asia. This El Niño was the biggest in the last 30 years, rivalling the well-known major El Niño in 1997/1998 (L'Heureux et al., 2016; Santoso et al., 2017). Page et al. (2002) estimated that the biomass burning in 1997 emitted a massive amount of carbon into the atmosphere, ranging between 810 and 2570 Tg C.

Compared to 1997, various observations were available in 2015, and several studies used those observations to estimate the fire-induced carbon emissions. Field et al. (2016) reported that the annual total carbon emission induced by the fires in 2015 was 380 Tg C, which was based on the Global Fire Emissions Database version 4s (GFED4s: Mu et al., 2011; Randerson et al., 2012; Giglio et al., 2013; van der Werf et al., 2017). The GFED4s data are derived from active fire data from the Moderate Resolution Imaging Spectroradiometers (MODIS) onboard the Terra and Aqua satellites. Huijnen et al. (2016) estimated the emission to be 289 Tg C by combining total column carbon monoxide (CO) data from the satellite-onboard instrument of Measurements of Pollution in the Troposphere (MOPITT) with emission factors estimated from local measurements of smoke. In their estimate, the fire-induced CO emission data from the Global Fire Assimilation System (GFAS v1.2: Kaiser et al., 2012) were modified to be consistent with the MOPITT CO observations, resulting in a downward shift from the original estimate of GFAS v1.2. Yin et al. (2016) also used the column CO data from MOPITT for estimating the carbon emission in Equatorial Asia. They used multi-tracer (CO, methane and formaldehyde) inverse analysis data (Yin et al. 2015) and estimated a fire-emitted CO of 122 Tg CO for 2015. With a prescribed ratio of the emission factors between total carbon and CO, this number leads to 510 Tg C for the total carbon emissions.

The total carbon emission estimates of the above studies were obtained from the fire-related data of MODIS and atmospheric CO mole fractions of MOPITT and not from observations of atmospheric $CO_2$, which is the major constituent of emitted carbon. Heymann et al. (2017) first used atmospheric $CO_2$ mole fraction data to estimate the fire-induced carbon emission in Equatorial Asia for 2015. They used the column-averaged dry-air mole fraction of $CO_2$ from the Orbiting Carbon Observatory-2 satellite (Crisp et al., 2008, 2015) and obtained a $CO_2$ emission estimate of 748 Mt $CO_2$ (equivalent to 204 Tg C) from July to November 2015, which covers the beginning and end of the fire season. Their estimate was 35% and 30% smaller than the MODIS-based emission estimates of GFED4s and GFAS v1.2, respectively. This lower estimate is more consistent with the estimate of Huijnen et al. (2016) than that of Yin et al. (2015).

Thus, the estimates of the fire-induced carbon emission in Equatorial Asia for 2015 are still uncertain, though they are consistently much smaller than that for 1997. As discussed by Field et al. (2009, 2016) and Yin et al. (2016), a nonlinear sensitivity of the fire emission to the climate conditions contributed to the notable discrepancy of the fire-emission amount between 1997 and 2015. However, the underlying mechanisms are unclear, and further investigation and a more accurate emission estimate are required. Importantly, the previous studies mainly relied on satellite data of atmospheric $CO_2$ or CO. These estimates have possible errors because satellite data are not well retrieved when there are smoke or clouds. Heavy smoke occurred from the fires in 2015 (Field et al., 2016). Furthermore, cumulus clouds reside over Equatorial Asia at high probability, although convective activity decreases during the dry season.



In this study, we estimated carbon emissions in Equatorial Asia for 2015 using in situ atmospheric observations by aircraft and ship. The observational data were obtained from the commercial aircraft observation project of Comprehensive Observation Network for TRace gases by AIrLiner (CONTRAIL: Machida et al., 2008) and the National Institute for Environmental Studies (NIES) Volunteer Observing Ship (VOS) Programme (Tohjima et al., 2005; Terao et al., 2011; Nakaoka et al., 2013; Nara et al., 2011, 2014, 2017). Because of in situ measurements, the observational data provide much higher

accuracy than the satellite observations used in previous studies. The moderate distance of the observational locations from the source areas (i.e., in the free troposphere or offshore) should ensure enough spatial representativeness of the observations in the inverse analysis. Given the sparse ground-based observations in Equatorial Asia, these programmes provide valuable opportunities to investigate the fire-induced emissions in the region. The long-term aircraft observation (the predecessor of CONTRAIL) observed $CO_2$ and CO mole fraction variations associated with El Niño over the western Pacific since 1993

(Matsueda et al., 2002, 2019). Its occasional observation flight to Singapore (Matsueda and Inoue, 1999) and a campaign flight over Australia and Indonesia (Sawa et al., 1999) captured pronounced elevations of CO from the Equatorial fires in 1997. Furthermore, Nara et al. (2017) observed prominent $CO_2$ and CO enhancements from the peatland fires in Equatorial Asia in 2013 by NIES VOS.

        To link the atmospheric observations to surface carbon fluxes, we performed an inverse analysis of atmospheric $CO_2$
using the NICAM-based Inverse Simulation for Monitoring $CO_2$ (NISMON-$CO_2$) (formerly NICAM-TM 4D-Var: Niwa et al., 2017a, 2017b). The inversion system uses the Nonhydrostatic Icosahedral Atmospheric Model (NICAM: Tomita and Satoh, 2004; Satoh et al., 2008, 2014)-based transport model (NICAM-TM: Niwa et al., 2011b). Using the same atmospheric transport model, Niwa et al. (2012) performed a $CO_2$ inverse analysis and demonstrated a strong constraint of the CONTRAIL data for Equatorial Asia. In this study, we estimated surface fluxes at a higher resolution using a more sophisticated inversion method

than that of Niwa et al. (2012), namely the four-dimensional variational (4D-Var) method (Niwa et al., 2017a). The 4D-Var estimates fluxes at a model grid-resolution to address flux signals from spatially limited phenomena such as biomass burning. We newly implemented CO into the inverse system to evaluate combustion sources. In our inverse analysis, we predominantly used atmospheric $CO_2$ observations from CONTRAIL and evaluated the inversion results using independent $CO_2$ and CO observations from NIES VOS. Finally, we performed an inverse analysis using both the CONTRAIL and NIES VOS $CO_2$

observations to enhance the reliability of the inverse analysis.

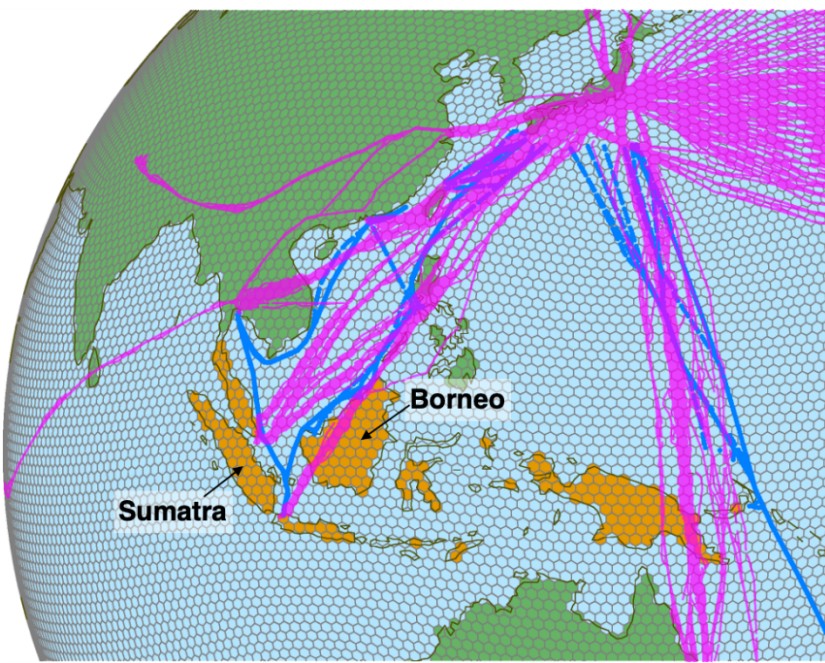

**Figure 1:** Locations of the observations obtained by CONTRAIL (magenta) and NIES VOS (blue) for Nov 2014–Jan 2016. Pentagons and hexagons in grey denote the icosahedral grids of NICAM (the grid interval is ~112 km), those filled in orange indicate Equatorial Asia, the target region of this study.

## 2 Methods

### 2.1 Observations

#### 2.1.1 CONTRAIL

The CONTRAIL data were obtained from in situ $CO_2$ measurements by Continuous $CO_2$ Measurement Equipment (CME), which is installed onboard the Boeing 777-200ER and -300ER of Japan Airlines (Machida et al., 2008; Sawa et al., 2012; Umezawa et al., 2018). For the analysis period from November 2014 to January 2016, the number of CONTRAIL-CME data exceeds 1.3 million, comprising 10-s interval data from ascending/descending sections and 1-min interval data from cruising sections. In the analysis, we only used data in the free troposphere, derived by excluding data in the stratosphere and the planetary boundary layer identified by thresholds of two potential vorticity units (PVU, 1 PVU = $10^{-6}$ m$^2$ s$^{-1}$ K kg$^{-1}$) and $Ri$ = 0.25 ($Ri$ is the bulk Richardson number), respectively (Sawa et al., 2008, 2012). This data filtering is needed because the signals of surface fluxes being efficiently attenuated in the stratosphere, and lower altitude data could be affected by local emissions from a neighbouring city of an airport (Umezawa et al., 2020). After filtering, the number of observations is still as large as 1.1 million. In particular, the observational coverage for Equatorial Asia is noteworthy, which is predominantly contributed by high-frequency flights between Japan and Singapore.



### 2.1.2 NIES VOS programme

The NIES VOS programme has been conducting atmospheric and surface ocean observations in the Pacific Ocean using commercial cargo vessels (Tohjima et al., 2005; Terao et al., 2011; Nakaoka et al., 2013; Nara et al., 2011, 2014, 2017). The observation network ranges from Japan to North America, Oceania (Australia and New Zealand), and Southeast and Equatorial

Asia. In 2015, the motor vessel named *Fujitrans World* (owned by the Kagoshima Senpaku Kaisha, Ltd. Kagoshima, Japan) was used for observations in Southeast and Equatorial Asia. Onboard the ship, an in situ measurement system continuously observed atmospheric mole fractions of greenhouse gases and other related atmospheric species (Nara et al., 2017). In this study, in addition to $CO_2$, atmospheric CO data were used for the proxy of fire-induced emissions. The ship normally travels once a month, but for 2015, observational data were obtained in January and from May to November. It takes approximately

two weeks to travel around Southeast and Equatorial Asia. In this study, we used 1-h interval data that passed careful quality control. Using ancillary data of the cruising speed and mole fractions of related species (e.g., ozone), the quality control excluded mole fraction data of $CO_2$ and CO that were judged as the ship's exhaust and contaminated by local ports.

### 2.2 Inverse analysis

### 2.2.1 Inversion system and transport model

Similar to previous inversions (e.g., Baker et al., 2006; Chevallier et al., 2010; Rödenbeck, 2005), the inverse analysis of this study is based on Bayesian estimation (e.g., Rayner et al., 1996; Enting, 2002). The cost function is defined as

$$J(\delta x) = \frac{1}{2}\delta x^{\mathrm{T}}\mathbf{B}^{-1}\delta x + \frac{1}{2}(M(x_0 + \delta x) - y)^{\mathrm{T}}\mathbf{R}^{-1}(M(x_0 + \delta x) - y), \tag{1}$$

where $\delta x$ is the control vector, including parameters to be optimised, $y$ represents the vector of observations, and $x_0$ denotes the basic model state of the parameters. The matrices $\mathbf{B}$ and $\mathbf{R}$ are the prescribed error covariance for $\delta x$ and the model-

observation mismatch, respectively. The operator $M(.)$ describes the forward simulation, including linear spatiotemporal interpolation to each observational location/time. In this inverse analysis, $x_0$ and $\delta x$ comprise prescribed surface $CO_2$ flux data and deviation from them, respectively, and the operator $M(.)$ represents the atmospheric transport. Atmospheric mole fraction observations of $CO_2$ are input to the vector $y$.

       In this study, we used the 4D-Var method to obtain an optimal vector $\delta \mathbf{x}$ that minimises the cost function. In the

method, an optimal parameter vector is sought by iterative calculations using the gradient of the cost function,

$$\nabla J_{\delta x} = \mathbf{B}^{-1}\delta x + \mathbf{M}^{\mathrm{T}}\mathbf{R}^{-1}(M(x_0 + \delta x) - y), \tag{2}$$

where $\mathbf{M}^{\mathrm{T}}$ is the transpose of the tangent linear operator $\mathbf{M}$ (in this study, $\mathbf{M}\delta x \approx M(\delta x)$ because of the linearity of the problem). The $\mathbf{M}^{\mathrm{T}}$ calculation requires an adjoint model.

       The inversion system NISMON is specifically designed for the inverse analysis of an atmospheric constituent (Niwa

et al., 2017a, 2017b). In the system, the forward model of NICAM-TM simulates atmospheric mole fractions from given surface fluxes, and its adjoint model calculates the sensitivities of fluxes against atmospheric mole fractions (Niwa et al.,





2017b). Specifically, the continuous adjoint model was chosen for the adjoint calculation, assuring monotonicity of tracer concentrations and sensitivities at the expense of minor nonlinearity (Niwa et al., 2017b). The optimisation calculation uses the quasi-Newton algorithm of the Preconditioned Optimizing Utility for Large-dimensional analyses (POpULar: Fujii and Kamachi, 2003; Fujii, 2005; Niwa et al., 2017a).

The atmospheric transport model NICAM-TM adopts an icosahedral grid system with hexagonal or pentagon-shaped grids (Fig. 1) that are produced by the recursive division of an icosahedron. All the model simulations were performed at a horizontal resolution of glevel-6 (*n* of glevel-*n* denotes the number of divisions of an icosahedron, representing the level of the model horizontal resolution). The averaged grid interval of glevel-6 is 112 km, sufficiently resolving the major archipelagos in Equatorial Asia (Fig. 1). For forward and adjoint simulations of atmospheric transport, archived meteorological data drive NICAM-TM, which is an off-line calculation. The meteorological data were prepared in advance from the simulation of the parent model NICAM, whose wind fields are nudged towards Japanese 55-year Reanalysis data (JRA-55: Kobayashi et al., 2015; Harada et al., 2016) (see Niwa et al., 2017b for a detailed description of the archived meteorological data). Other model settings can be found in Niwa et al. (2017b).

### 2.2.2 Implementation of CO

In this study, we newly implemented a CO function in the above inversion system to use CO as a proxy of fire-induced emissions. It also considers oxidation from CO to $CO_2$, which could have measurable effects on $CO_2$ observations near fires. Figure 2 shows a schematic diagram for the forward and adjoint simulations of NICAM-TM, including CO. This CO function considers only the chemical reaction with hydroxyl radicals (OH). The OH fields are given as input data and, hence, the model does not have nonlinear chemical reactions, retaining the linearity, which is assumed in the inverse analysis theory. Furthermore, the oxidation from methane ($CH_4$) to CO with OH is also considered. For simplicity, however, the atmospheric mole fraction of $CH_4$ was set at a globally constant value of 1844 ppb ($= 10^{-9}$ mol$^{-1}$), which was derived from the global annual mean mole fraction for 2015, reported by the World Data Center for Greenhouse Gases (WDCGG: WMO, 2018). The atmospheric three-dimensional data of OH were derived from the TransCom-$CH_4$ project (Patra et al., 2011). In the model, the contribution of oxidation from biogenic volatile organic compounds (BVOCs) to CO is given as emissions from the earth's surface. Note that we did not input CO observations to the inverse analysis. The dominant part of the observations was from the in situ CONTRAIL measurements, which did not have simultaneous CO data available. In the inversion, the CO flux in the model was modified along with the biomass burning emission of $CO_2$, as described in the next section.

### 2.2.3 Flux model

As described in Fig. 2, we introduced scaling factors to surface fluxes, which is another updated feature of the inversion system from Niwa et al. (2017a). The surface $CO_2$ flux input to the model, $f_{CO_2}$, is described as

$$f_{CO_2}(x,t) = \big(1 + \Delta a_{fos}(x,t)\big) f_{fos}(x,t) - \big(1 + \Delta a_{GPP}(x,t)\big) f_{GPP}(x,t)$$



$$+\big(1 + \Delta a_{\mathrm{RE}}(x,t)\big)f_{\mathrm{RE}}(x,t) + \big(1 + \Delta a_{\mathrm{fire}}(x,t)\big)f_{\mathrm{fire}}(x,t)$$

$$+f_{\mathrm{ocn}}(x,t) + \Delta f_{\mathrm{ocn}}(x,t), \tag{3}$$

where $x$ and $t$ indicate flux location and time, and $f$ represents prescribed flux data, whose subscripts of fos, GPP, RE, fire and

ocn denote flux components of fossil fuel combustion and cement production, Gross Primary Production (GPP) and respiration

(RE) of terrestrial biosphere, biomass burning and ocean, respectively. Note that a positive value indicates a flux towards the

atmosphere. Each flux component data could have different temporal resolutions (e.g., monthly, daily), and flux values are

linearly interpolated in time to each model time step. Datasets used for each flux component are described in the following

section. Their coefficients of $\Delta a_{\mathrm{fos}}$, $\Delta a_{\mathrm{GPP}}$, $\Delta a_{\mathrm{RE}}$ and $\Delta a_{\mathrm{fire}}$ are modification scaling factors for corresponding flux components,

of which the values could be varied at each model grid. We did not apply the scaling factor to the ocean flux but introduced

the deviation of the prescribed flux $\Delta f_{\mathrm{ocn}}$ because the ocean flux has both negative and positive values and its spatiotemporal

flux phase could not be modified when introducing a scaling factor. Note that the other flux components should have all

positive values. The phases of spatiotemporal variations of terrestrial biosphere flux (e.g., seasonal cycle) could be modified

because GPP and RE are separately optimised. The above modification scaling factors and $\Delta f_{\mathrm{ocn}}$ were the parameters to be

optimised in the inverse analysis.

For the surface CO flux, we considered fossil fuel, BVOCs and biomass burning emissions. For the biomass burning

emissions, we imposed the common scaling factor with that of $CO_2$. Therefore, in the inversion, the biomass burning emission

of CO was modified along with that of $CO_2$. The modification of the biomass burning emission could also be made by signals

transported via atmospheric CO that should have oxidised to $CO_2$ (Fig. 2).

In Eq. (3), the temporal resolution of a flux-scaling factor could be different from that of its corresponding flux and

be different by regions (Table 1). In this study, we set the daily temporal resolution for the scaling factors of GPP, RE and

biomass burning emissions in Equatorial Asia so that the inversion could exploit the full information of those surface fluxes

from the observations. For the rest of the region, we set monthly temporal resolutions. For the scaling factor of fossil fuel

emissions, we set an annual temporal resolution for Equatorial Asia, and we did not optimise the flux for the rest of the region,

i.e., the modification factor was set to 0. We set the monthly temporal resolution for the deviation of the ocean flux.

**Table 1:** Temporal resolution and standard error and error correlation of each flux component, which was separately configured for
Equatorial Asia and the rest of the world. Note that the ocean flux was optimised by its absolute value and the others by their scaling factors;
therefore, the monthly standard deviation (SD) of the long-term data was used for the ocean flux error and ratios were used for the other flux
errors.

| Flux component | Temporal resolution | | Standard error | | Error correlation (space/time) | |
|---|---|---|---|---|---|---|
| | Eq. Asia | Rest | Eq. Asia | Rest | Eq. Asia | Rest |
| fos | Annual | N/A | 10% | N/A | None/None | N/A |
| GPP, RE | Daily | Monthly | 40% | 10% | None/3 day | 1000 km/None |
| fire | Daily | Monthly | 80% | 100% | None/3 day | None/None |
| ocn | N/A | Monthly | N/A | SD | N/A | 3000 km |


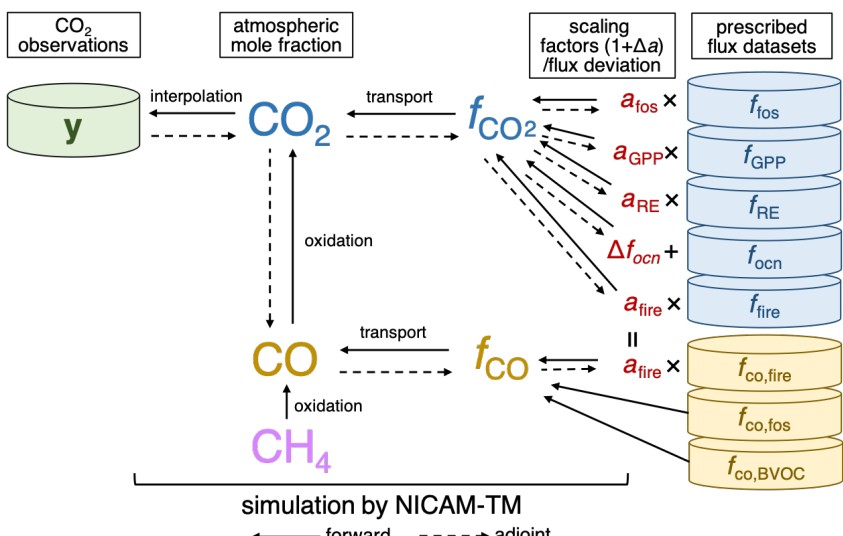

**Figure 2:** Schematic diagram of the $CO_2$–CO forward/adjoint calculations in NICAM-TM

### 2.2.4 Error covariance matrices

As in the case of the different flux temporal resolutions, we constructed the flux error covariance matrix **B** for Equatorial Asia and the rest of the world separately. Table 1 summarises the standard errors and error correlations that were introduced into the diagonal and off-diagonal elements of **B**, respectively. For GPP and RE, we assume 40% error for daily fluxes in Equatorial Asia and 10% for monthly fluxes in the rest of the world, which means 0.16 and 0.01 for the diagonal elements of **B**. The higher standard error for Equatorial Asia allows observations to modify surface fluxes sufficiently. Nevertheless, a 3-day scale temporal error correlation was introduced to stabilise flux estimates. The smaller standard error for the rest of the world had to constrain the surface flux to the prior because we did not use enough observations to cover the globe. Furthermore, for the stabilisation, the spatial error correlation with the 1000 km scale length was introduced. The above error correlations were defined by the Gaussian function (Niwa et al., 2017a). Similarly, the errors for fossil fuel and fire emissions were introduced, but without error correlations. For the monthly ocean flux errors, we used the standard deviation of the long-term data (1990–2016) and introduced the spatial error correlation of 3000 km. Table 1 shows each parameter.

### 2.2.5 Prescribed flux dataset

For $f_{fos}$ and $f_{ocn}$ in Eq. (3), we used monthly mean data of fossil fuel and cement production emission from the Carbon Dioxide Information Analysis Center (CDIAC) (Andres et al., 2016) and of air-sea $CO_2$ flux from the Japan Meteorological Agency


(Takatani et al. 2014; Iida et al., 2015), respectively. For $f_{GPP}$ and $f_{RE}$, we used 3-hourly data for resolving distinct diurnal cycles of terrestrial biosphere flux. These $f_{GPP}$ and $f_{RE}$ were originally based on monthly mean data from the Carnegie-Ames-Stanford Approach (CASA) model (Randerson et al., 1997), but were modified according to the inversion of Niwa et al. (2012).

They were further downscaled in time to 3-hourly with 2 m temperature and downward shortwave radiation data of JRA-55 using Olsen and Randerson's (2004) method.

In the inversion of Niwa et al. (2012), the classical low-resolution inversion method (e.g., Enting, 2002) was used, in which the global terrestrial area was divided into 31 regions, and the scaling factors for those regions were optimised. Furthermore, the inversion used both surface and CONTRAIL data for 2006–2008, and the mean flux data of those three years

were used in this study. Therefore, such an integrated flux could produce consistent atmospheric mole factions with the observations from the surface to the upper troposphere, although some discrepancies could arise because of the different analysis period. In this study, those discrepancies were modified using the consistent year data of CONTRAIL and further flux information was exploited because of using the 4D-Var high-resolution (model grid level) inversion, with a specific focus on Equatorial Asia.

For the biomass burning flux of $f_{fire}$, we used four datasets and performed independent inversions to evaluate sensitivities to the biomass burning data (Table 2). The first is the mean of the GFED4s and GFAS v1.2 (noted as GG). The second and third ones are from GFED4s (GD) and GFAS v1.2 (GS), respectively. The fourth one is made by excluding emissions in Equatorial Asia from GG (NO). For NO, we replaced the biomass burning term of Eq. (3) by $\left(0 + \Delta a_{fire}(x, t)\right) f_{fire}(x, t)$ in Equatorial Asia, where $f_{fire}$ is the same as GG.

For CO, the same biomass burning datasets from GFED4s and GFAS v1.2 were used. The rest of the CO fluxes from fossil fuel use and oxidation of BVOCs were derived from the Emission Database for Global Atmospheric Research (EDGAR) version 4.3.2 (Janssens-Maenhout et al., 2019) and the process-based model of terrestrial ecosystems, the Vegetation Integrative SImulator for Trace gases (VISIT: Ito and Inatomi, 2012; Ito, 2019), respectively. For EDGAR v4.3.2, we used emission data of 2012 (the latest data available) for the simulation of 2015.


**Table 2:** Observation and prior fire-emission data for each inverse analysis experiment

| Experiment name | Observation | Fire prior |
|---|---|---|
| C_GG | CONTRAIL | (GFAS + GFED)/2 |
| C_GD | CONTRAIL | GFED |
| C_GS | CONTRAIL | GFAS |
| C_NO | CONTRAIL | No fire in Equatorial Asia |
| CV_GG | CONTRAIL, VOS | (GFAS + GFED)/2 |



### 2.2.6 Initial mole fraction field and analysis period

In addition to the flux-scaling factors, the model parameter vector includes the global offset of atmospheric mole fractions. Therefore, $\delta\boldsymbol{x}$ of Eqs. (1) and (2) is constructed as

$$\delta\boldsymbol{x} = (\Delta\boldsymbol{a}, \Delta\boldsymbol{f}_{\mathrm{ocn}}, \Delta c)^{\mathrm{T}}, \tag{4}$$

where $\Delta\boldsymbol{a}$ and $\Delta\boldsymbol{f}_{\mathrm{ocn}}$ represent all the modification scaling factors and ocean flux deviations of Eq. (3), respectively, and $\Delta c$ denotes the modification to the global offset. Thus, its corresponding basic state vector $\boldsymbol{x}_0$ is described as $\boldsymbol{x}_0 =$

$(1, ..., 1, \boldsymbol{f}_{\mathrm{ocn}}, 0)^{\mathrm{T}}$. Note that the forward model calculation started from a reasonable spatial gradient, which was prepared in advance by a spin-up calculation. At the beginning of the 4D-Var iterative calculation, all the elements of $\delta\mathbf{x}$ were set to zero as the initial estimates.

The target period for this study is the whole year of 2015. However, in the inverse calculation, two extra months were added before the target period to attenuate the errors in the initial mole fraction fields before the beginning of 2015, which was

inevitable because of the global unique parameter described above ($\Delta c$). Furthermore, one more month was also added after the target period to well constrain fluxes at the end of 2015. Therefore, the inverse calculation period consists of 15 months from November 2014 to January 2016.

### 2.3 Notation of sensitivity tests

As described in 2.2.5, we performed inversion analyses with four biomass burning datasets (GG, GD, GS and NO). We only

used the CONTRAIL data, but with GG, we additionally performed an inversion using the NIES VOS data and CONTRAIL to leverage all available observations, denoting C_ and CV_ as prefixes, respectively. Thus, we have five inversion results (C_GG, C_GD, C_GS, C_NO and CV_GG) (Table 2). Note that although they used different biomass burning data, prior flux errors for biomass burning (Table 1) were commonly used. It is true even in C_NO whose practical prior uncertainty in Equatorial Asia is 80% of GG.

## 3 Results

In this section, we first describe spatiotemporal features of the CONTRAIL and NIES VOS observational data with supplemental model analyses. Then, we show the inversion results and demonstrate their validity by comparing posterior mole fractions of CO and $CO_2$ with the NIES VOS data.

### 3.1 Observational features

As shown in Fig. 3, the CONTRAIL aircraft frequently flew to Singapore 209 times during 2015. These high-frequency observations manifested a small but distinct seasonal cycle of $CO_2$ mole fractions around Equatorial Asia, with double peaks in April–May and December, and a minimum during June–October, depending on altitudes and latitudes. Furthermore, the CONTRAIL aircraft frequently observed additional highly elevated mole fractions below 3 km altitude over Singapore (Fig.





3a, lower panel), which could be attributable to local or regional emissions in Equatorial Asia. The model with the prior flux

data produced similar mole fraction elevations; however, their timing and magnitudes were sometimes different from the

observations (Fig. 3b). A further model analysis with prior biomass burning data suggested that fire contributions to the

observed mole fraction elevations were limited mostly within the latter period of the dry season from mid-August to the

beginning of November (Fig. 4b). In the other seasons, the model showed almost no contributions from fire emissions (not

shown). In particular, the model showed a distinct fire contribution at the end of September, which elevated mole fractions up

to the upper troposphere by ~4 ppm. In the observations, although similar mole fraction elevations are found in the upper

troposphere, its magnitude is smaller (~2 ppm). Furthermore, the observation shows a slightly later peak that lasted until the

beginning of October (Fig. 4a). After that, the observations also captured elevated mole fraction events from mid-October

onward. However, the prior model estimate showed smaller fire contributions in October than in September (Fig. 4b), although

the simulated total $CO_2$ mole fractions were comparable to the observations (Fig. 3b), indicating that non-fire emissions (e.g.,

from terrestrial biosphere respiration and fossil fuel emission) had a certain level of contribution during this period.

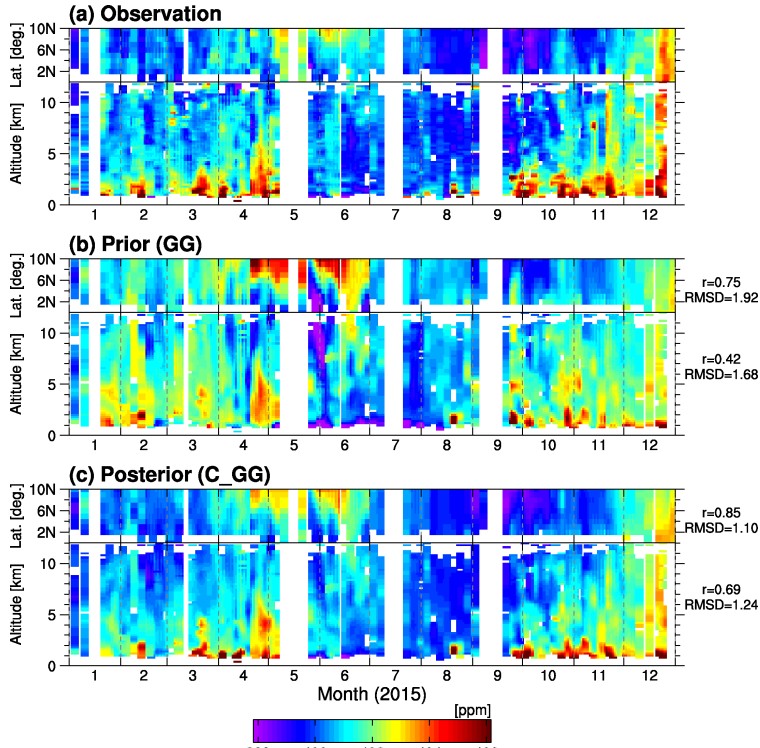

**Figure 3:** $CO_2$ mole fractions in the free troposphere around Equatorial Asia (note that data in the boundary layer and stratosphere are excluded (see the main text)) observed by CONTRAIL (a) and their corresponding model values from prior (GG) (b) and posterior (C_GG) fluxes (c). Each upper panel presents a time-latitude cross-section from cruising mode data (~11 km above sea level) within the longitude
range of 90°–130°E, and the lower one shows a time-altitude cross-section from ascending/descending data over Singapore. Note that the data in the upper panels are not only from the Singapore flights but from all flights within the range. For visualisation, data are all 5-day running mean. Note also that an additional offset of 1.93 ppm is added to prior mole fractions so that the resulting global offset equals the posterior one. On the right-hand side, correlation coefficients and root-mean-square difference (RMSD) (ppm) between the simulated and observed mole fractions are noted for each time-latitude and time-altitude cross-section.


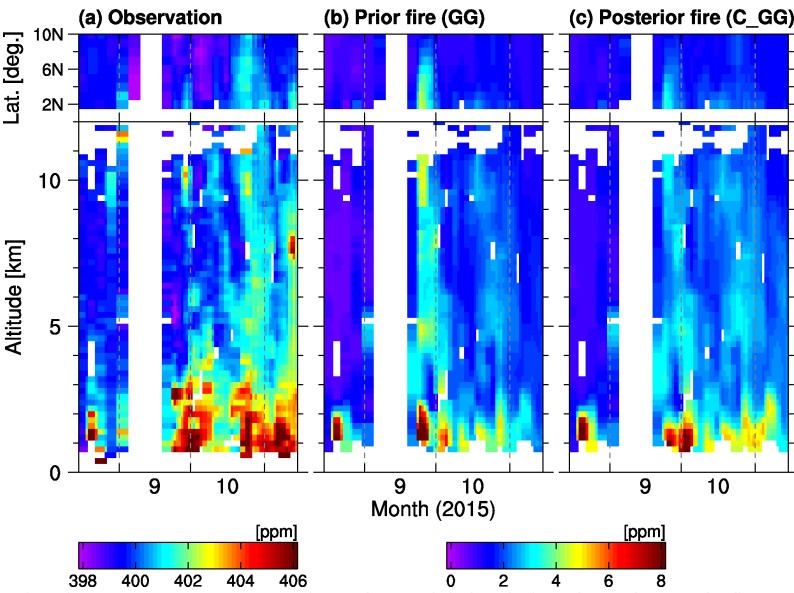

**Figure 4:** The same as Fig. 3, but for 15 August–15 November. The model simulations here show only fire contributions.

Figure 5 shows $CO_2$ and CO mole fractions observed along the track of the NIES VOS around Equatorial Asia. The
NIES VOS observations in both September and October 2015 captured coincident elevations of $CO_2$ and CO mole fractions
in the east of the Malay Peninsula and west of Borneo. By performing a transport simulation of tagged fire-induced CO tracers
(the fire emission of GG was used here), we found that the fires in Borneo and Sumatra contributed almost every notable mole
fraction elevation, except for 17 September and 15–16 October, both of which might be contributed by the fossil fuel emission
in Jakarta city (Fig. 6). As highlighted by the grey shades in Fig. 6, seven of those events contributed by the fires are designated
by P1, P2,…, P7 in this study. These events will be used for evaluating the inverse analysis, especially for fire-emission
features, as described in Section 3.2.2. Note that mole fraction data during 13–15 September and 11–14 October were excluded
before the analysis. During these periods, some data were not correctly obtained because the signals were too large and out of
the measurable range. Furthermore, the NIES VOS ship travelled slowly or stayed around the Malacca Straits, resulting in
contamination by the ship-self exhaust.



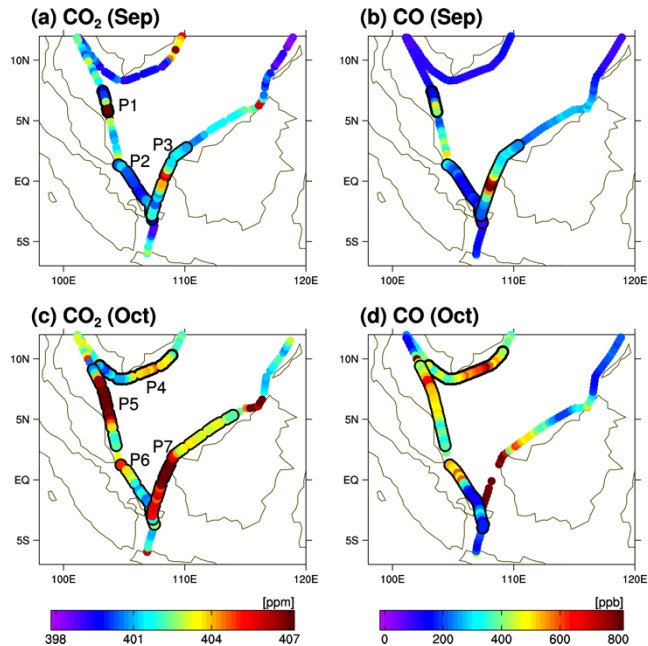


**Figure 5:** Mole fractions of $CO_2$ (left) and CO (right) along the cruise tracks of NIES VOS for September (upper) and October (lower) 2015. Data enclosed by black lines with P# represent designated fire-induced high mole fraction events (see also Fig. 6).

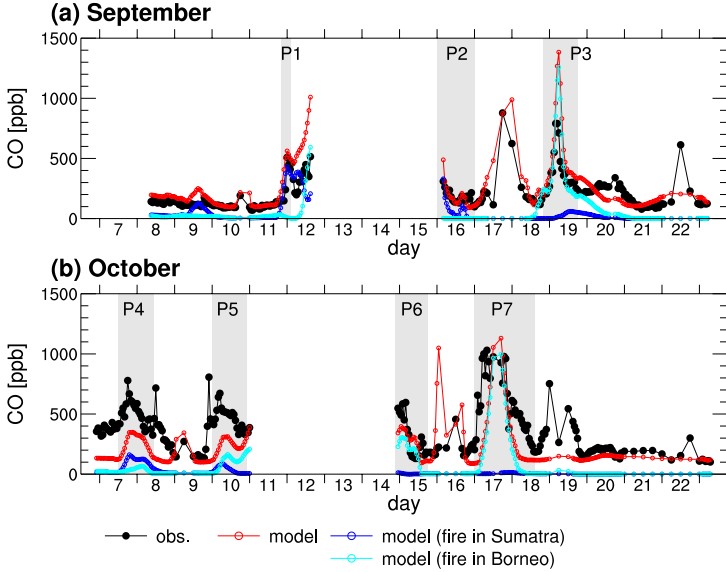


**Figure 6:** Time series of CO mole fractions obtained by the in situ NIES VOS measurement (black) for September (a) and October (b) and corresponding simulation results by NICAM-TM with prior CO emission data (red). Model simulations only from fire emissions in Sumatra and Borneo are also denoted by blue and cyan colours, respectively. Grey shades with P# indicate the fire-induced elevated mole fraction events.





Figure 7 shows sensitivities of the CONTRAIL and NIES VOS observations against surface $CO_2$ fluxes, i.e., footprints, for September and October 2015, calculated by the adjoint model of NICAM-TM (Niwa et al., 2017b). The CONTRAIL footprints represent sensitivities of observations obtained during ascending or descending flights over Singapore (i.e., the data shown in the lower panel of Fig. 4a). For both September and October, the calculated footprints indicate that the CONTRAIL observations could provide significant constraints on flux estimates for Equatorial Asia, especially Borneo (Figs.
7a and b). These widespread footprint features are because the data were obtained in the free troposphere, which is an advantage of aircraft observations in terms of representativeness. Figure 7 also suggests that the constraint is stronger during October than September because the number of data is larger in October (Fig. 3 or Fig. 4). Compared to CONTRAIL, the NIES VOS footprints are restricted to the ocean (Figs. 7c and d) because the observations were made within a marine boundary layer. It would also be contributed by weak wind fields, which are typical in the tropics. Nevertheless, there are some sensitivities of
the NIES VOS observations on the coasts of the islands, with which the significantly large fire emissions elevated the mole fractions of atmospheric $CO_2$ and CO (Figs. 5 and 6).

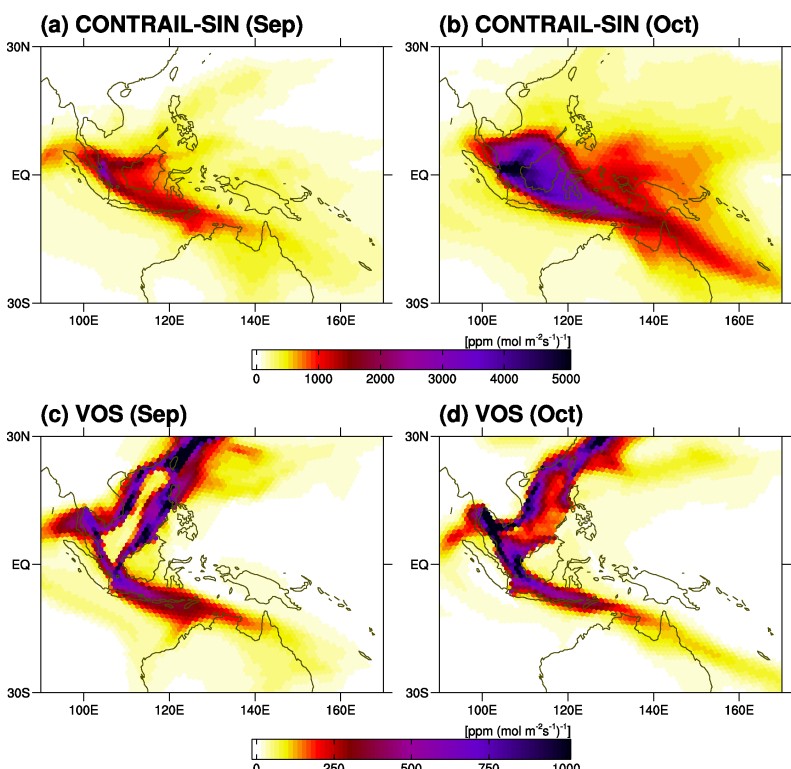

**Figure 7:** Sensitivity of surface $CO_2$ flux against the observations (e.g., footprints) of CONTRAIL over Singapore (upper) and NIES VOS (lower) for September (left) and October (right) 2015.






### 3.2 Inversion results

### 3.2.1 Posterior fluxes

In this study, we investigate surface fluxes by the sum of $CO_2$ and $CO$ fluxes, defined as a carbon flux. We do not consider $CH_4$ fluxes, although they could contribute some percentage of the total carbon flux (Huijnen et al., 2016). Furthermore, we

evaluate the carbon flux separately for the total net flux and biomass burning emission. Note that the total net flux includes terrestrial biosphere fluxes, biomass burnings emissions and fossil fuel emissions.

Table 3 summarises the total net and fire carbon fluxes of Equatorial Asia estimated by the five sets of inversions. The prior biomass burning emissions of GG, GD and GS are consistently 300 Tg C for September–October, which constitutes ~80% of the annual total fire emission and amounts to more than 80% of the total net flux we prescribed as the prior (355–360

Tg C) for September–October. By inversion, all experiments, other than C_NO, estimated smaller total net fluxes by ~10% (304–324 Tg C) than the priors, and they were mostly contributed by the smaller estimates of fire emissions (256–277 Tg C). Interestingly, even when prior fire emissions were excluded in Equatorial Asia (C_NO), a significantly large fire emission of 122 Tg C was retrieved for September–October, indicating that the CONTRAIL data have fire-emission signals. However, the estimate is half of the others, indicating some dependency of the inversion on the prior fire emissions.


**Table 3:** Total net flux and fire emission of carbon from Equatorial Asia for September–October 2015. Figure 1 defines the geographical region of Equatorial Asia. Note that the total net flux includes terrestrial biosphere fluxes, biomass burnings emissions and fossil fuel emissions. Annual flux values for 2015 are also noted in parentheses. The prior fluxes with the four biomass burning emissions are presented, as well as the posterior fluxes of the five inversions.

|  | Total net flux [Tg C] | Fire emission [Tg C] |
| --- | --- | --- |
| Prior (GG) | 357 (677) | 299 (388) |
| Prior (GD) | 360 (685) | 301 (396) |
| Prior (GS) | 355 (669) | 296 (379) |
| Prior (NO) | 59 (289) | 0 (0) |
| C_GG | 324 (613) | 277 (363) |
| C_GD | 304 (598) | 256 (343) |
| C_GS | 320 (604) | 265 (348) |
| C_NO | 211 (451) | 122 (131) |
| CV_GG | 322 (608) | 273 (362) |


Figures 8 and 9 show the $CO_2$ flux distributions for September and October, respectively. Here, we present the posterior fluxes of C_GG and C_NO only, but those of the other inversions show almost similar distribution features to C_GG. In September, C_GG estimated significantly high emissions in southeast Sumatra and south of Borneo, where the fire emission dominated in the prior flux (Fig. 8), supporting prior knowledge of biomass burnings. However, the total emission estimate is





smaller than the prior by 31 Tg (Fig. 8c). In October, the differences between prior and posterior fluxes of C_GG were moderate, with a net difference of 2.7 Tg (Fig. 9c). This small change in October indicates that the simulated mole fractions from the prior flux were well consistent with the CONTRAIL observations.

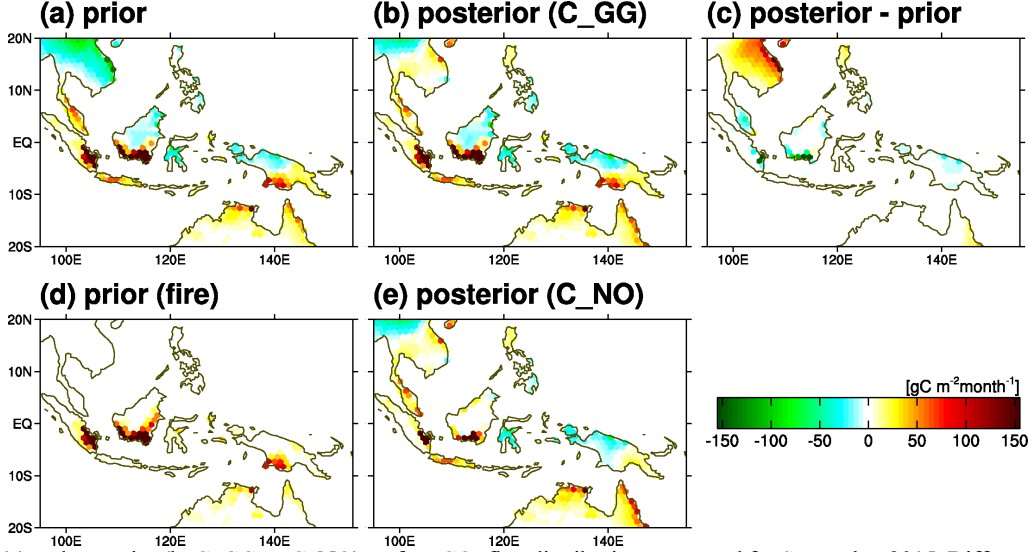

**Figure 8:** Prior (a) and posterior (b: C_GG, e: C_NO) surface $CO_2$ flux distributions averaged for September 2015. Difference between prior and posterior fluxes (c) and prior fire emissions (d) are also shown.

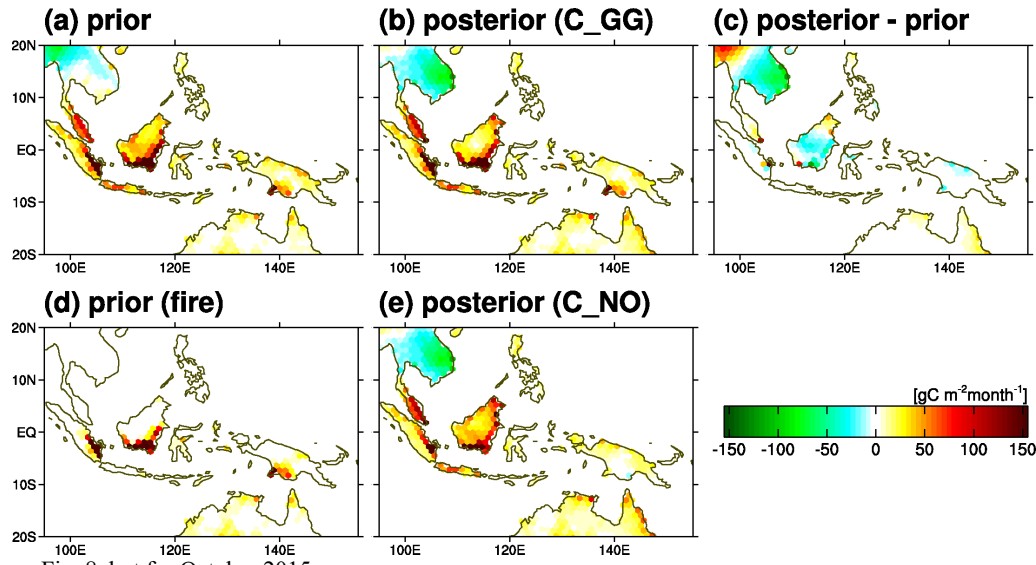

**Figure 9:** Same as Fig. 8, but for October 2015.






As shown in Table 3, even when the fire emission was excluded from the prior, the inversion estimated notable fire emissions (C_NO), in both September and October (Figs. 8e and 9e). Furthermore, the locations of the estimated fire emissions are well coincident with those of prior fire emissions. They were to some extent guided by the higher prior flux errors that were derived from the fire-emission data (the uncertainty was set as 80% of GG), which was confirmed by another sensitivity

test without prior uncertainty of fire emissions (not shown). Nevertheless, this result confirms that the CONTRAIL data have information about biomass burning emissions.

Figure 10 shows temporal variations of the total net carbon flux in Equatorial Asia. Here, the posterior fluxes of C_GG and CV_GG are shown (panel a) and the difference from each prior is presented as Δ (panel b). Note that the time series of Δ is smoother because Δ is the parameter optimised by the inversion with a 3-day temporal correlation scale. This temporal

correlation works as a smoother. The differences between the two posterior fluxes are marginal, indicating a limited effect of adding the NIES VOS data to the CONTRAIL data because the number of CONTRAIL data is overwhelming and the footprint of CONTRAIL covers Equatorial Asia much more extensively in space (Fig. 7). Compared to the prior flux, the posterior fluxes have a smaller peak at the beginning of September, whereas they show larger peaks from the end of September to the beginning of October. In the latter part of October, the prior and posterior fluxes are similar.


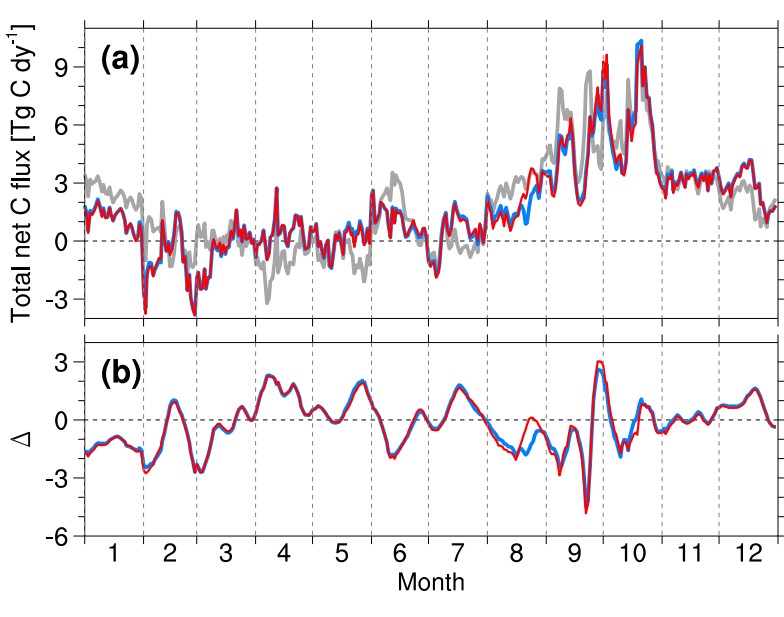

**Figure 10:** Time series of posterior total net carbon fluxes (a) and their differences from prior (Δ) (b) for 2015. Posterior fluxes of C_GG (blue) and CV_GG (red) and their prior flux (grey) are presented.

Figure 11 exhibits temporal variations of fire emissions during the fire season. In Equatorial Asia, although the timings of the emission peaks presented by GFED and GFAS are coincident with each other, their magnitudes are significantly

different. GFED has larger peaks than GFAS in September, whereas it has smaller ones in October (Fig.11a). Generally, the posterior fire emissions, except for C_NO, fall within the range of both prior emission estimates. In September, the posterior estimates are consistent and their magnitudes are closer to GFAS rather than GFED. In October, however, notable

discrepancies occur among the posterior emissions, which is contributed by different emission estimates for Sumatra rather than Borneo according to the breakdown of the posterior estimates (Figs. 11b and c). In October, GFAS has higher emissions than GFED on both islands and its degree is more prominent in Sumatra. These different prior fire-emission estimates might have contributed to the large discrepancy among posterior estimates.

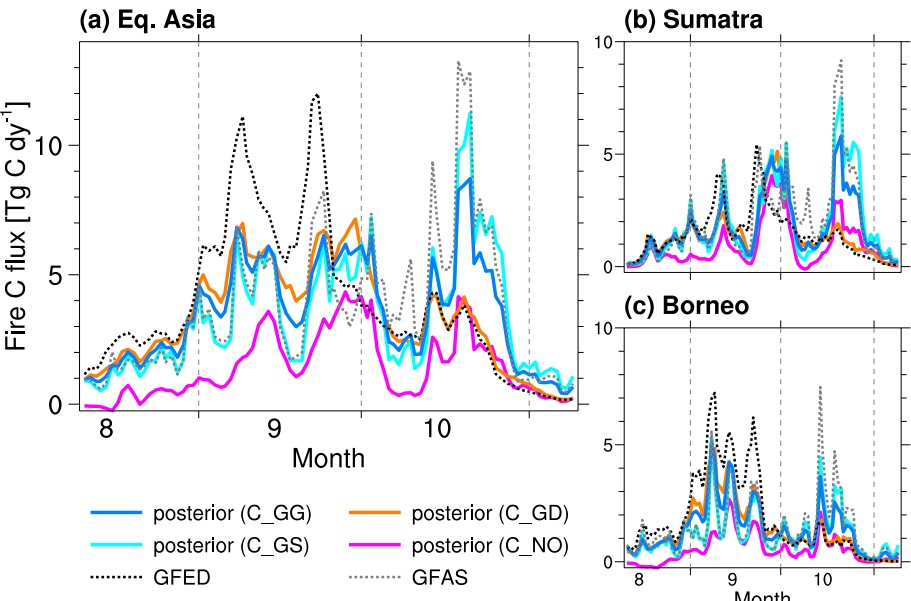

**Figure 11:** Time series of posterior and prior fire carbon emissions over the fire season (mainly, September–October) of 2015. The posterior fluxes of C_GG (blue solid), C_GD (orange solid), C_GS (cyan solid), and C_NO (magenta solid) are shown. Prior data of GFED (black dotted) and GFAS (grey dotted) are also shown.


As shown in Fig. 7, the CONTRAIL footprint covers Borneo better than Sumatra, and its degree is larger in October than in September. In practice, however, the constraint on fire emission has a different feature, as seen in the noticeable spread of fire-emission estimates in October (Fig. 11c). Note that the spread of estimates, including C_NO, is small in Sumatra at the end of September (Fig. 11b). In this period, the observational data and model analysis indicated that strong fire signals reached Singapore, although its timing, as suggested by the model, was slightly earlier (Fig. 4b). For this event, the inversion

successfully optimised the fire emission in Sumatra with strong constraints by the observations, even from the no-fire prior (C_NO), resulting in the later-shifted peak with the consistent magnitude of 4–5 Tg C dy$^{-1}$ (Fig. 11b).



### 3.2.2 Posterior mole fractions

In this section, we evaluate the simulated atmospheric $CO_2$ and CO mole fractions from the posterior fluxes. First, as shown in Fig. 3, the posterior mole fractions of $CO_2$ simulated for CONTRAIL have shown much better agreement with the observations than the prior ones, demonstrating that the inverse analyses were reasonably well performed. Compared to the simulation results of the prior fluxes, the posterior mole fractions have greater correlation coefficients and smaller root-mean-square differences from the observations (see the numbers at the right-hand side of Fig. 3). In the following, we will compare the model with the $CO_2$ and CO observations of NIES VOS, which were left independent of the inversions, except for CV_GG. As demonstrated by Fig. 2, the posterior CO flux includes the fire emission modified according to the modification of $CO_2$ fire emissions. To elucidate carbon fluxes in Equatorial Asia, the NIES VOS data used here are limited in the neighbouring region (95°–125°E and 10°S–15°N).

In the comparative analysis of $CO_2$ observations, an additional offset of 1.93 ppm is added to the prior $CO_2$ mole fractions so that the resulting global offset becomes equivalent to the posterior ones, i.e., $\Delta c$ of Eq. (4) is 1.93 ppm (note that there is almost no difference in the global offset among the five inversions). Because the initial global offset was arbitrarily given, the comparison analysis of $CO_2$ should exclude the effect of the improvement in the global offset to better understand the inversion effects.

Figure 12 demonstrates how the posterior mole fractions of $CO_2$ and CO were improved from the prior ones. Comparing the posterior results with the prior ones, we found better consistency with the NIES VOS observation, which is true for both $CO_2$ and CO. Especially, its degree is notable for September; all inversions, except CV_GG, reduced the root-mean-square difference (RMSD) of $CO_2$ from 2.14–2.62 ppm to 2.05–2.09 ppm, whereas those of CO were reduced from 92–211 ppb to 80–111 ppb. These results indicate the validity of the inversions that used the CONTRAIL $CO_2$ observations. Especially, the experiment without the prior fire emission (C_NO) remarkably improved the correlation coefficients of $CO_2$ and CO for both September and October (in September from 0.43 to 0.56 for $CO_2$ and from 0.34 to 0.81 for CO, whereas in October from 0.29 to 0.49 for $CO_2$ and from −0.18 to 0.49 for CO); however, the other inversions, except for CV_GG, did not improve the correlation coefficients significantly. In both months, CV_GG showed the best scores for $CO_2$, which is reasonable because it used the $CO_2$ observations of NIES VOS. Note, however, that CV_GG used only the $CO_2$ observations, but not those of CO. Therefore, the RMSD reduction of CO for September by CV_GG (from 136 to 103 ppb) demonstrates some improvement in fire emissions. Therefore, it is better to use both the CONTRAIL and NIES VOS observations for flux estimations; however, the impact of NIES VOS is limited for the total carbon fluxes in Equatorial Asia (Table 3 and Fig. 10).

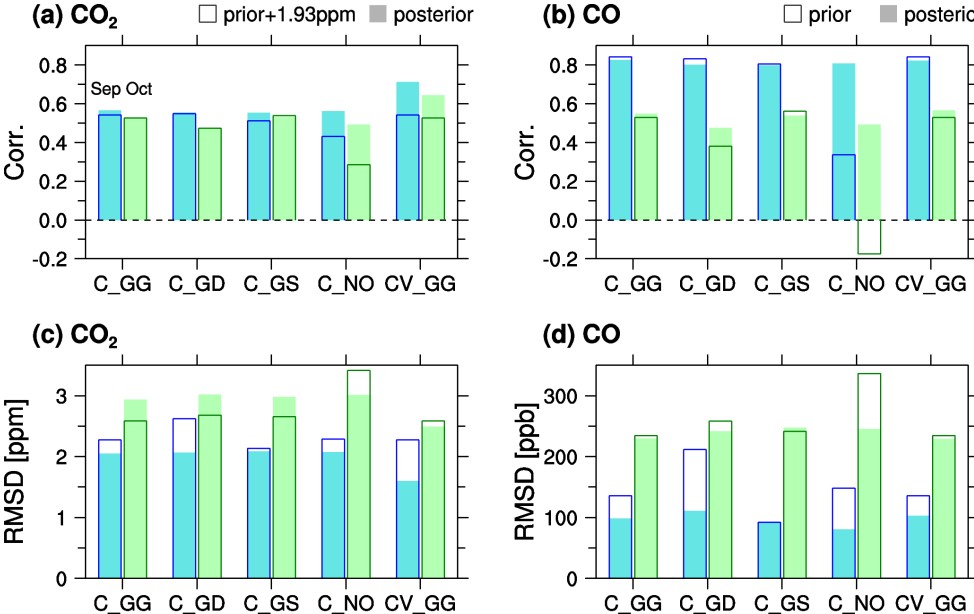

**Figure 12:** Correlation coefficient (upper panels) and root-mean-square-difference (RMSD) (lower panels) between the observed and simulated NIES VOS $CO_2$ (left) and CO (right) mole fractions. The NIES VOS data used here are limited within the range of 95°–125°E and 10°S–15°N for September (light blue) and October (light green) 2015. The model simulations are derived from each posterior flux (C_GG, C_GD, C_GS, C_NO, and CV_GG) (filled bar) and its corresponding prior flux (open bar). To consider the initial global offset error, we added 1.93 ppm to every prior value so that the prior initial global offset becomes equivalent to the posterior one.

For each elevated mole fraction event defined by Figs. 5 and 6, we calculated an enhancement ratio of $\Delta CO/\Delta CO_2$ from the reduced major axis regression, as Nara et al. (2017) did. For all peaks, except P1 and P2, every correlation between $CO_2$ and CO variations is statistically significant ($p < 0.05$) for both the observations and simulations. For P1 and P2, some simulated results could not have significant correlations; therefore, we combined the two events so that every correlation was statistically significant (combining P1 and P2 would be reasonable because the Sumatra fires contributed to both; see Fig. 6). The enhancement ratios for those events could provide implications for emission ratios between CO and $CO_2$. Note that the simulated ratios are derived from the posterior fluxes, but the overall feature does not change when the prior fluxes are used because the fire-emission ratios between CO and $CO_2$ were unchanged by the inversion. Figure 13 depicts the observed and simulated enhancement ratios and the observed correlation coefficients for each event. The figure shows that the observed ratio has a significantly large variation from 0.034 ppb/ppb to 0.169 ppb/ppb. Interestingly, the higher ratios were obtained from events that were likely contributed by the Borneo fires (P3, P6 and P7; see also Fig. 6). The model with any posterior flux reproduced a pattern similar to the observed one. However, for the events from September to early October (P1 and P2, P3 and P4), the model coherently overestimated the enhancement ratios. The model and observations respectively show 0.067–0.104 ppb/ppb and 0.038 ppb/ppb for P1 and P2, 0.139–0.165 ppb/ppb and 0.117 ppb/ppb for P3, and 0.115–0.133 ppb/ppb and 0.066 ppb/ppb for P4. In the latter events, the simulated ratios have substantial spreads among the different inversions,





especially for P6 (0.080–0.145 ppb/ppb) and P7 (0.123–0.182 ppb/ppb), and the observed ratios are almost at the highest level

of the simulated spreads. A further discussion is made in the following section.


**Figure 13:** Observed and simulated enhancement ratios of ΔCO/ΔCO₂ (solid line) and observed correlation coefficients between ΔCO and ΔCO₂ (grey bar) for each elevated mole fraction event defined by Figs. 5 and 6. The observed enhancement ratios are coloured in black. The simulated values were derived from posterior CO and CO₂ fluxes of C_GG (magenta), C_GD (red), C_GS (blue), C_NO (green), and 475 CV_GG (cyan).

## 4. Discussion

In this study, we extensively used the aircraft observations from CONTRAIL to constrain carbon fluxes in Equatorial Asia,

focusing on the devastating fire event in 2015. With the help of NIES VOS observations, especially its CO data, we have

demonstrated the validity of our inverse analysis. The estimates of biomass burning emissions are moderately smaller than the

prescribed datasets of GFED and GFAS. Our conceivably best estimate of CV_GG, which used both the CONTRAIL and

NIES VOS data, amounts to 273 Tg C and 362 Tg C for fire-induced carbon emissions during September–October and all

months in 2015, respectively. These numbers are in better agreement with previous top-down estimates of Huijnen et al. (2016)

(227 Tg C for September–October, and 289 Tg C for the annual total) and Heymann al. (2017) (204 Tg C for July–November)

than that of Yin et al. (2016) (510 Tg C for the annual total). Furthermore, the fire-induced carbon emission of 273 Tg C for

September–October is also consistent with an aerosol-based study of Kiely et al. (2019), the best estimate of which is 247 Tg

C as the sum of CO₂ and CO emissions for Equatorial Asia but not including eastern areas (e.g., Papua New Guinea).





This study used high-precision in situ observations of $CO_2$ to estimate carbon fluxes in contrast to studies that used satellite observations of CO (Huijnen et al., 2016; Yin et al., 2016). Therefore, we obtained the total net carbon flux that

included biomass burning emission, terrestrial biosphere photosynthesis and respiration, and fossil fuel emission. The estimated total net carbon flux amounts to 322 Tg C for September–October (CV_GG), 85% of which is contributed by fire emissions (Table 3), indicating that flux variations of terrestrial photosynthesis and respiration under severe drought were not as large as those of the biomass burning emissions in 2015. This result indicates that biomass burning emission is the main driving force of interannual variations of carbon fluxes in Equatorial Asia, which is a unique feature of the carbon flux in this

region, compared with other tropical regions. Carbon fluxes in the tropics are considered to have significant sensitivities to climate variations, especially to El Niño, with major driving forces of terrestrial biosphere flux changes in response to temperature and precipitation changes (Yang and Wang, 2000; Zeng et al., 2005; Wang et al., 2013).

As described in Section 2.2, we used the common scaling factor for $CO_2$ and CO fire fluxes; therefore, the ratio of the emission factors (CO/$CO_2$) was fixed to that of prior fire-emission data (GFED and GFAS) and burned carbon mass was

modified by the inverse analysis. It is likely that the spatial pattern of fire-emission ratios was reasonably represented because the model reproduced the observed variation in the enhancement ratio that might be caused by the difference in their origins (Figs. 6 and 13). However, as shown in Fig. 13, the model overestimated the enhancement ratios from September to early October, irrespective of its origin (Borneo or Sumatra). Meanwhile, it was not the case for the latter period, although the simulated ranges were large, indicating that the temporal change of the fire-emission ratio, i.e., a decrease of combustion

efficiency, might not be well represented in the fire-emission data. Typical fire-emission ratios of CO/$CO_2$ are 0.2–0.3 mol/mol for peatlands and 0.1 mol/mol for tropical forests, respectively (Akagi et al., 2011; Huijnen et al. 2016; Stockwell et al., 2016). Therefore, the observed smaller enhancement ratio infers that the contribution of fires (or smouldering) in dried peatlands was smaller in the early fire period than expected. This might have partially resulted in the smaller-than-prior estimates of the fire-induced carbon emission (Table 3) because peatlands have high carbon density and are dominant sources of carbon (Page et

al., 2002). Nevertheless, the uncertainty is high, as demonstrated by the model large spreads, especially for the latter period, and we need more observations for robust estimations of the fire-emission ratio.

We noted that the inverse analysis of this study has several limitations. First, we employed only $CO_2$ observations in the inverse analysis, which does not allow us to distinguish biomass burnings from other terrestrial fluxes. To separate fire-induced and other terrestrial fluxes, it would be helpful to incorporate CO observations simultaneously with $CO_2$ observations

into the inverse analysis; however, the availability of in situ CO observations (Novelli et al., 2003) is limited, especially for Equatorial Asia. A joint $CO_2$–CO inversion is left for a future study.

The second limitation is the dependency of the inverse analysis results on the prior estimate. We found that our inverse calculations had a significant sensitivity to prior fire-emission data. The posterior fluxes were similar when GFED or GFAS data were used as the prior. However, when the prior fire emissions were excluded (i.e., C_NO), the posterior flux had much

lower values, indicating that we cannot fully constrain the fluxes with CONTRAIL alone and that prior fire-emission data should be as accurate as possible. The relatively small sensitivity to the difference between GFED and GFAS was because the



original GFED and GFAS data are comparable. Nevertheless, the well-constrained flux at the end of September in Sumatra (Fig. 11) tells us that we could obtain a sufficient constraint from CONTRAIL when the time and location of the observations coincide with the timing of airflow rich in emission signals. Such an airflow dependency could be reduced by making the observations denser in space and time. To this end, the CONTRAIL project continues effort to reduce chances of missing data.


Finally, model transport errors would be the third limitation. Figures 3 and 4 show that even after the inversion, the model could not sufficiently reproduce high mole fractions near the surface, suggesting some limitation of the model. Compared to the wind data obtained from the CONTRAIL aircraft, the speed and direction of winds over Singapore were well simulated in the model (not shown). Therefore, representative errors that include both model transport and fluxes could be one cause. A higher resolution model is desirable, and is left for a future study with advanced computational resources.


In this study, we tried a regionally focused inversion using different flux parameter settings for Equatorial Asia and the rest of the world (Table 1), which gave a sufficient degree of freedom to fluxes in Equatorial Asia, while strongly constraining fluxes to the prior ones in the rest of the world. This inversion approach would be acceptable only when prior fluxes can produce comparable spatiotemporal variations of atmospheric $CO_2$ with observations, which was confirmed using the previous inversion flux of Niwa et al. (2012). Furthermore, Niwa et al. (2012) found that CONTRAIL data could independently constrain the fluxes in Equatorial Asia itself, supporting the validity of separating fluxes in Equatorial Asia from those in the other regions. Nevertheless, the inversion flux used as the prior in this study was optimised for the different years, which introduced a certain level of uncertainty. In future analysis, an effort for reducing uncertainties will be made by performing a global inverse analysis by combining ground-based stations and CONTRAIL data.


## 5. Conclusions


In this study, an inverse system was developed to estimate high-spatiotemporal resolution fluxes in a focused area and incorporate CO as a proxy for combustion sources. We performed the inverse analysis for carbon fluxes in Equatorial Asia during the historic El Niño of 2015.

In contrast to many studies that used aircraft data for evaluating inversion results as independent data (e.g., Chevallier et al. 2019), we extensively used the CONTRAIL aircraft data in the inverse analysis and demonstrated that the aircraft data could constrain flux estimates efficiently. It is essential for Equatorial Asia because there are insufficient ground-based observations in the region. Furthermore, the upward airflow, which is typical in the tropics, makes it difficult for remote ground-based stations to capture flux signals (Niwa et al., 2012).


We estimated the fire-induced carbon flux to be 273 Tg C for September–October. This number accounts for 75% of the annual fire emission and 45% of the annual net carbon flux in Equatorial Asia, demonstrating that fire emission is a major driving force of the carbon flux in the region. Although the inversions have a certain degree of sensitivity to prior fire-emission data, they coherently estimated smaller amounts than the prescribed biomass burning data. One cause could be that peatland fires were not as severe in 2015 as expected, as suggested by the model overestimation of the enhancement ratios $\Delta CO/\Delta CO_2$ captured by the NIES VOS observation. Nevertheless, this study is compatible with previous studies because a significantly


smaller amount of carbon was released in 2015 than in 1997, of which the El Niño intensity was comparable to that of 2015. This could be because the intensity of land-use change has decreased in recent decades (Kondo et al., 2018). Another possible underlying mechanism is the difference in precipitation patterns and amounts between 2015 and 1997 (Fanin and van der Werf, 2017). However, further investigation is needed by combining process-based terrestrial biosphere models, biomass burning emission models and inverse estimates, as in this study.

560       Although it is smaller than 1997, our estimate of the fire-induced emissions in 2015 is still notably high. Field et al. (2016) pointed out that the fire emissions estimated by GFED for Equatorial Asia in 2015 are higher than the annual fossil fuel emissions of Japan. Our estimate is smaller than that of GFED, but still comparable to the latest Japanese inventory (338 TgC yr$^{-1}$ for 2018 (GIO and MOE, 2020)). Using an atmospheric climate model, Shiogama et al. (2020) projected that Equatorial Asia would experience stronger droughts than that in 2015 in a future warmer climate condition. To reduce fire-induced carbon
emissions from such drought events, chances of ignition must be reduced. The continuous monitoring of carbon emissions with high-precision atmospheric observations is indispensable for mitigation measures against fires. The ongoing activity of aircraft- and ship-based observations used in this study will continue to provide practical implications and reliable quantitative estimates of carbon emissions from Equatorial Asia along with an inverse analysis.

**Data Availability**

The $CO_2$ mole fraction data of CONTRAIL (Machida et al., 2018) and NIES VOS used in this study are available from the Global Environmental Database (GED) of NIES (http://db.cger.nies.go.jp/portal/geds/atmosphericAndOceanicMonitoring). The CO observational data of NIES VOS are available on request to H. Nara.

**Author Contributions**

YN designed and conducted the inversion analyses. TM, YS, HM, YN and TU conducted the CONTRAIL observations and HN, SN, HT and YT conducted the NIES VOS observations. AI provided the VISIT data. YN prepared the manuscript with contributions from all co-authors.

**Competing Interest**

The authors declare that they have no conflict of interest.

**Acknowledgements**



This study was supported mainly by the Environment Research and Technology Development Fund of the Ministry of the Environment, Japan, and the Environmental Restoration and Conservation Agency of Japan (JPMEERF20142001 and JPMEERF20172001), whose project leader was Nobuko Saigusa of NIES. The work was also supported by the JSPS KAKENHI, Grant Number 19K03976. The inverse simulations in this study were performed on the NIES supercomputer system (NEC SX-ACE). The observations from the CONTRAIL project are conducted under great supports of Japan Airlines, JAMCO and the JAL Foundation. The observational projects of CONTRAIL and NIES VOS are financially supported by the research fund by Global Environmental Research Coordination System of the Ministry of the Environment, Japan (E1253, E1652, E1851). YN is grateful to the NICAM developers of the University of Tokyo, JAMSTEC, RIKEN and NIES for maintaining and developing NICAM. Our appreciation is also extended to Tomoko Shirai and Yoko Fukuda of NIES for developing and maintaining GED, through which we make our observational data publicly available.

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
