# Peer review of "Estimation of fire-induced carbon emission from Equatorial Asia in 2015 by using in situ aircraft and ship observations"

_Atmospheric Chemistry and Physics, 2020_

## Referee Comment (RC1) · Anonymous Referee #1 · 25 Feb 2021

This paper analyses $CO_2$ (and CO) flux anomalies from the Southeast Asia region, using a data assimilation method for estimation of $CO_2$ fluxes at transport model grid resolution. They have used unique observational datasets and a model framework. The region has already been widely studied by various groups and this paper would be a nice addition to the growing list of literature. My judgement is that the authors demonstrate a capacity that would be utilised in near-real-time in the future or on regular basis. The paper is generally well written and can be accepted for publication after a major revision, in my opinion. Please see my specific comments below for revisions, which I hope are useful for you to consider.

[Figure]

line 33: should you say 2nd strongest - please check MEI or something like that?

Section 2.1: Just to make sure, you haven't use any other observations apart from the CONTRAIL and VOS in this study. Could you be explicit please.

line 187: a bit of details or a reference is needed for the BVOC model here, which ones are considered etc. and how CO prodection is modelled.

Table 1: GPP, RE errors are quite large, which is nice. why is the error for fire emission is set to greater over the rest, compared to eq. asia?

line 221: CDIAC data are used for what time period ?

line 241: how were they derived ? did you model BVOC oxidation ?

line 258: should work in principle because you are starting with an inversion flux, but not sure 2-month of spin-up is sufficient

Figure 3: Are these plots better shown by "Model - Observation" for the lower 2 panels?

Figure 4: I wonder if you really need Figure 4

Figure 6: Why not show the CO2 time series as well, as that is the main focus of the paper?

line 343: Delete "We do not consider CH4...". I think, none will as ask for CH4 at this point.

line 352: since you are putting less prior flux uncertainties outside the domain of this analysis, this is bound to happen. do you have another simulation with uniform treatment of prior flux uncertainty ?

line 365: One of the problems here is the over interpretations of the flux differences, without giving a range of uncertainty e.g., how significant are 31 Tg vs 2.7 Tg mentioned

Figure 8 & 9: please mention in the caption if the prior flux map is same for both the

inversions (C_CG & C_NO). and also show the posterior - prior for the C_NO case. hopefully you will find a place to show the colour bar (on the right?)

line 378: Is this statement true? see over the Papua New Guinea or the southern part of Borneo, where are there are large differences in patterns

Figure 11: Nicer to have a common x-axis for these plots for easier reading.

line 410: not clear what you mean by "its degree"?

Figure 12: A bit surprised to see posterior RMSDs are quite often exceeding the prior values for October

Figure 13 and associated text: I am not very sure of usefulness of this discussion. You need to take in to account the ageing of the air mass, i.e., the influence of near and far fields in constructing the emission ratios. It is possible to implement such an analysis using trajectory model for instance.

Section 4: not sure if you need these detailed discussions, a table of fire emission estimation can be given in the results section or here for improving readability. Then some of the text, say on the modelling limitations, could be integrated with the conclusions or new subsection in Results

Section 5: Conclusions can focus on the outcome of this paper, with minimal or no referencing to old papers. Some can be kept for the significance of the paper, e.g., the ENSO related flux variability, which has been discussed in the results section.

---

## Referee Comment (RC2) · Julia Marshall (Referee) · 30 Mar 2021

The authors present a clear and well-structured inverse modelling study focusing on the application of in-situ measurements from aircraft and ships to target fluxes in Tropical Asia associated with the 2015 ENSO event. The focus is on biomass burning emissions, and the result is a reduction in the fire flux emissions of $CO_2$ compared to established satellite-based emission products, which is in line with other studies examining the same period and region but based on completely different data streams.

The topic is interesting and relevant, and it presents a good application of aircraft- (and ship-)based measurements for fluxes, and not just model validation. The figures are clear and well-designed to support the story of the paper, and the references to related studies are fully appropriate. As such, I would consider it appropriate for publication in ACP after some minor concerns have been addressed.

L159-164: I had some questions about the treatment of OH and CH4 here. Rather than assuming a constant value of CH4 everywhere, it might be more reasonable to use a fixed distribution, as the CH4 decreases with altitude. This decrease is most dramatic above the tropopause, which is not considered explicitly here, but is seen throughout the atmosphere, as the sources of CH4 are all located at the surface and the sink is (overwhelmingly) in the atmosphere. Because the WDCGG estimate is based on surface measurements, this will be an overestimation for an atmosphere-wide value. However, the amount of CO2 being created from the oxidation of CH4 is quite minor and as such it doesn't really matter much in this study – it was more a comment for future work. Regarding the OH: here I guess the Spivakovsky fields distributed within the TransCom CH4 project, reduced by 8%, are meant? If so, a reference to Spivakovsky et al. (1990) should be added.

I guess that these signals are really small anyhow, but it would good to quantify this, also for BVOCs: approximately how large are the different components in the simulated (prior) signal of CO2? What is essentially background (from outside of your targeted region), how much is from local anthropogenic emissions, how much from oxidation of other species, how much from the biosphere (net), and how much from biomass burning? I would be really interested to see the contribution of the different signals to the total simulated signal, even just for e.g. one simulated flight.

I also had a question about Figure 8 and 9 that I would like to see addressed in the text: what do you think is the reason for the large positive adjustment over the Indochinese Peninsula (Vietnam, Cambodia, Laos...) for this period? This is not seen in the fire prior at all, and I presume that it is a biosphere signal, but the adjustment is really large for both months, but in opposite directions. Indeed, it almost looks as if the prior for October better fits the posterior for September, and the prior for September the posterior for October. Is this a shifting in the seasonality of the biospheric fluxes from the climatological norm (or the 2003-2005 mean) due to El Nino? Some discussion of this would be welcome.

Finally, no estimate of the uncertainty of the resultant fluxes is presented, unless one considers the spread related to the different priors. Is such an estimate feasible with the inversions system presented here? Even if such a calculation is not technically feasible, some discussion of this shortcoming should be included. And one last minor point: I think it would be beneficial to explain earlier in the text (in the methods section) how the emission ratio of CO/CO2 is set. References to Akagi et al. only come much later (in the discussion).

Besides this, I made some language corrections/suggestions in the document itself, which I have uploaded. I hope that some of these may be helpful.

Please also note the supplement to this comment:
https://acp.copernicus.org/preprints/acp-2020-1239/acp-2020-1239-RC2-supplement.pdf

**Supplement:**

[revised manuscript text omitted]

---

## Author Comment (AC1) · 5 May 2021

Reply to Reviewer #1

This paper analyses CO2 (and CO) flux anomalies from the Southeast Asia region, using a data assimilation method for estimation of CO2 fluxes at transport model grid resolution. They have used unique observational datasets and a model framework. The region has already been widely studied by various groups and this paper would be a nice addition to the growing list of literature. My judgement is that the authors demonstrate a capacity that would be utilised in near-real-time in the future or on regular basis. The paper is generally well written and can be accepted for publication after a major revision, in my opinion. Please see my specific comments below for revisions, which I hope are useful for you to consider.

We really appreciate the reviewer for taking valuable time to review our paper and giving us useful comments and suggestions. Described below are our replies to the reviewer's comments with page and line numbers of the revised manuscript. The supplementary manuscript shows changes from the previous one.

line 33: should you say 2nd strongest - please check MEI or something like that?

We appreciate the reviewer for pointing it out. We rather intended it to be "one of the biggest", because the El Niño intensity differs with different indices.
We modified it to "This was one of biggest El Niño events". **[Line 33]**

Section 2.1: Just to make sure, you haven't use any other observations apart from the CONTRAIL and VOS in this study. Could you be explicit please.

We appreciate the reviewer's suggestion for clarification. Accordingly, we added the following sentence in the beginning of Section 2.1.
"In this inverse analysis, we only used the CONTRAIL and NIES VOS data, because they are predominant in the area we focused on. Here, we briefly describe those observations and further information can be found in the literatures cited therein." **[Lines 99-100]**

line 187: a bit of details or a reference is needed for the BVOC model here, which ones are considered etc. and how CO prodection is modelled.

First of all, we apologize for incorrect descriptions of the BVOC data of VISIT caused by misunderstanding among co-authors. We now realized that we did not consider the oxidation of BVOC, but only considered the direct emissions of CO from vegetation. Although we recognize that the oxidation of BVOC is important for simulating atmospheric CO, we think it would not significantly affect our results and conclusions in this study, because we performed the CO2-only inversion and focused only on biomass burnings. In this regard, we modified the descriptions of BVOC in the manuscript and replaced "$f_{CO,BVOC}$" with "$f_{CO,vege}$" in Fig. 2. In the VISIT simulation, CO emissions are estimated with the scheme of Guenther (1997) but using an emission factor by Tao and Jain (2005). The emission rates are estimated by light, temperature, vegetation leaf amount and seasonality, and the scheme has been calibrated with observational data.

We modified the last part of Section 2.2.2 as:

"In the model, the contribution of oxidation from biogenic volatile organic compounds (BVOCs) to CO is not considered yet, but direct CO emissions from vegetation are given at the earth's surface. Although the oxidations of $CH_4$ and BVOCs are significant sources of atmospheric CO, we treated

the former very simply and did not consider the latter. Therefore, we did not input CO observations to the inverse analysis. In the inversion, the biomass burning emissions of CO, which were predominant in Equatorial Asia, were modified along with those of $CO_2$, as described in the next section." **[Lines 167-171]**

Table 1: GPP, RE errors are quite large, which is nice. why is the error for fire emission is set to greater over the rest, compared to eq. asia?

In fact, the effective error of the prior fire is larger than 80%. This is because the posterior error correlations are considered in time (3 day), which introduces off-diagonal elements in **B** and inflates the errors in Equatorial Asia; that is not the case for the rest of the world. Error numbers summarized in Table 1 were determined after tens of preliminary experiments.

In order to elaborate the flux error settings, we modified the text as follows:

"Similarly, the errors for fossil fuel and fire emissions were introduced, but without error correlations."

=>

"Similarly, 80% and 100% errors for fire emissions were introduced for Equatorial Asia and the rest of the world, respectively, but without spatial error correlations. Note that the fire errors for Equatorial Asia are practically larger than 80%, because the 3-day temporal correlation inflates the errors. We put a 10% error on the fossil fuel emissions in Equatorial Asia." **[Lines 218-221]**

line 221: CDIAC data are used for what time period ?

We added the sentence describing how we made the fossil fuel data for 2015.

"Here, the fossil fuel emission data for 2015 were produced from the latest gridded CDIAC data for 2013 by scaling with the global total value for 2015 that is preliminarily reported by Le Quéré et al. (2015)." **[Lines 226-228]**

line 241: how were they derived ? did you model BVOC oxidation ?

Please see our reply to your query to line 187. For the CO emissions of VISIT, we added sentences as follows.

"In the VISIT simulation, CO emissions are estimated with the scheme of Guenther (1997), but using an emission factor by Tao and Jain (2005). The emission rates are estimated by light, temperature, vegetation leaf amount and seasonality, and the scheme has been calibrated with observational data." **[Lines 250-253]**

line 258: should work in principle because you are starting with an inversion flux, but not sure 2-month of spin-up is sufficient

We consider that the two-month is enough for the spin-up, because the initial mole fraction fields were prepared also by the inversion flux of Niwa et al. (2012). To make it clearer, we added the sentences below.

"Nevertheless, the initial mole fraction fields are consistent with observations to some extent, as they were prepared by the inversion flux of Niwa et al. (2012). This makes the two-month inversion spin-up reasonable." **[Lines 270-272]**

Figure 3: Are these plots better shown by "Model - Observation" for the lower 2 panels?

We agree with the suggestion and modified Fig. 3 and its caption accordingly.

Figure 4: I wonder if you really need Figure 4

We think Figure 4 is useful for demonstrating probable contribution of the fire emissions to the CO2 mole fractions observed by CONTRAIL. As shown by Figs. 3 and 4, the observed CO2 do not show fire-related variations so clearly, Figs 4b and 4c help us to imagine how large the fire contributions are.

Figure 6: Why not show the CO2 time series as well, as that is the main focus of the paper?

As pointed out, we did not show the CO2 time series in Fig. 6. That is because we have already shown both the CO2 and CO observations of NIES VOS in Fig. 5 and from those we can easily recognize the two species are correlated with each other. The aim of Fig. 6 is to investigate if those correlated mole fraction peaks are contributed by fires or not, and, if so, which fire (Sumatra or Borneo) contributed to them. However, we realized that the data ranges of Figs. 5 and 6 were not exactly the same; that of Fig. 5 showed a slightly smaller range (up to 12°N and 120°E) than that of Fig. 6 (up to 15°N and 125°E). Therefore, we expanded the area of Fig. 5 slightly so that its data range is consistent with that of Fig. 6. Furthermore, we added the text "These time series depict the data limited within the range of 95°-125°E and 10°S-15°N consistently with Figs. 5b and d." in the caption of Fig. 6.

line 343: Delete "We do not consider CH4...". I think, none will as ask for CH4 at this point.

Accordingly, we deleted it.

line 352: since you are putting less prior flux uncertainties outside the domain of this analysis, this is bound to happen. do you have another simulation with uniform treatment of prior flux uncertainty ?

As pointed out, this result has some dependency on the prior uncertainties, and actually we put less uncertainties, i.e. strong constraint, on prior fluxes in the outside region. However, their flux contributions to atmospheric CO2 mole fractions could be big because the area is significantly larger than the target region. Therefore, we do not consider that the small flux uncertainty of the outside region does not necessarily induce the large fire emission estimate in the target region. Furthermore, we tested no fire-emission-based uncertainty, which is almost similar to the case of "uniform treatment of prior flux uncertainty", and we obtained the similar number for the fire emissions in Equatorial Asia (this is written in the manuscript **[Lines 403-405]**).

line 365: One of the problems here is the over interpretations of the flux differences, without giving a range of uncertainty e.g., how significant are 31 Tg vs 2.7 Tg mentioned

We consider that these numbers between the prior and posterior fluxes are not considerably large. In order to clarify this, we modified the text as follows.

"This small change in October indicates that the simulated mole fractions from the prior flux were well consistent with the CONTRAIL observations."

=>

"In fact, these flux changes are small compared with the differences between GFED and GFAS (GFED minus GFAS is 92 Tg C and -87 Tg for September and October, respectively), indicating that the simulated mole fractions from the prior flux of C_GG were overall consistent with the CONTRAIL observations." **[Lines 390-392]**

Figure 8 & 9: please mention in the caption if the prior flux map is same for both the inversions (C_CG & C_NO). and also show the posterior - prior for the C_NO case. hopefully you will find a place to show the colour bar (on the right?)

According to the suggestion, we added the text

"Note that the prior estimate of (a) was used both for C_GG and C_NO, while the prior fire estimate of (d) was used only for C_GG."

in the caption of Fig. 8. Furthermore, we modified Figs. 8 and 9 to include the posterior – prior for the C_NO case.

line 378: Is this statement true? see over the Papua New Guinea or the southern part of Borneo, where are there are large differences in patterns

As pointed out, the pattern is different for Papua New Guinea, but we consider that the pattern is similar to each other for southeast Sumatra and south of Borneo, where the fire was most intense. To limit the statement valid only for the Sumatra and Borne regions, we added text as follows.

"… are well coincident with those of prior fire emissions in southeast Sumatra and south of Borneo." **[Line 403]**

Figure 11: Nicer to have a common x-axis for these plots for easier reading. line 410: not clear what you mean by "its degree"?

We revised Fig. 11 accordingly. Besides, we changed "its degree" to "the sensitivity" for clarification. **[Line 435]** Please note that footprint quantifies a sensitivity of an observation against surface fluxes.

Figure 12: A bit surprised to see posterior RMSDs are quite often exceeding the prior values for October

Because the NIES VOS is independent of the inversions except for CV_GG, the RMSD should not be necessarily reduced. This is probably because some representative errors in surface fluxes or atmospheric transport were larger in October than in September, which can be inferred from the larger RMSDs and the smaller correlations. We added this discussion in the revised manuscript as follows.

"Meanwhile, the RMSDs of both $CO_2$ and CO were not necessarily reduced for October. This is attributable to insufficient representativness of surface fluxes or atmospheric transport, which can be inferred from the larger RMSDs and the smaller correlations in October than those of September. Nevertheless, the experiment without the prior fire emission (C_NO) exhibited smaller RMSDs with

the posterior $CO_2$ and CO fluxes for both September and October. Furthermore, the improvement in the correlation coefficients is remarkable…" **[Lines 461-466]**

Figure 13 and associated text: I am not very sure of usefulness of this discussion. You need to take in to account the ageing of the air mass, i.e., the influence of near and far fields in constructing the emission ratios. It is possible to implement such an analysis using trajectory model for instance.

We did not intend to infer the absolute value of the emission ratio by Fig. 13. As the reviewer says, estimating the absolute value of the emission ratio is difficult due to the aging of the airmass. However, we believe that we could infer if the emission ratio prescribed in the flux data is larger than real or not, by comparing the simulated ratios with those of the observation, in which the aging of the airmass is fairly considered by the atmospheric transport model.

Section 4: not sure if you need these detailed discussions, a table of fire emission estimation can be given in the results section or here for improving readability. Then some of the text, say on the modelling limitations, could be integrated with the conclusions or new subsection in Results

Following the reviewer's suggestion, we modified the Discussion section. Particularly, we moved a part of the first paragraph to Section 3.2.1 that discusses the obtained fire emissions as shown by Table 3 comparing with other study results. **[Lines 369-375]** Furthermore, we also moved the sentence "with the help of NIES VOS observations, especially its CO data, we demonstrated the validity of our inverse analysis" to the Conclusion section. **[Lines 568-569]**

Meanwhile, we keep the discussion of the modeling limitations, because it is not only the output of the inversion but also includes some perspectives, which we consider suitable for Discussion.

Section 5: Conclusions can focus on the outcome of this paper, with minimal or no referencing to old papers. Some can be kept for the significance of the paper, e.g., the ENSO related flux variability, which has been discussed in the results section.

We have tried to reduce the volume of this section. First, we deleted the sentence referring Niwa et al. (2012). Furthermore, we reduced the last paragraph by moving the sentence "Fields et al. (2016) pointed out … for 2018 (GIO and MOE, 2020)" to Section 3.2.1 **[Lines 375-378]**.

---

## Author Comment (AC2) · 5 May 2021

Reply to Julia Marshall (Reviewer #2)

The authors present a clear and well-structured inverse modelling study focusing on the application of in-situ measurements from aircraft and ships to target fluxes in Tropical Asia associated with the 2015 ENSO event. The focus is on biomass burning emissions, and the result is a reduction in the fire flux emissions of CO2 compared to established satellite-based emission products, which is in line with other studies examining the same period and region but based on completely different data streams.

The topic is interesting and relevant, and it presents a good application of aircraft- (and ship-)based measurements for fluxes, and not just model validation. The figures are clear and well-designed to support the story of the paper, and the references to related studies are fully appropriate. As such, I would consider it appropriate for publication in ACP after some minor concerns have been addressed.

We are grateful to Julia Marshall for taking valuable time to review our manuscript and giving positive comments. Described below are our replies to the reviewer's comments with page and line numbers of the revised manuscript. The supplementary manuscript shows changes from the previous one.

L159-164: I had some questions about the treatment of OH and CH4 here. Rather than assuming a constant value of CH4 everywhere, it might be more reasonable to use a fixed distribution, as the CH4 decreases with altitude. This decrease is most dramatic above the tropopause, which is not considered explicitly here, but is seen throughout the atmosphere, as the sources of CH4 are all located at the surface and the sink is (overwhelmingly) in the atmosphere. Because the WDCGG estimate is based on surface measurements, this will be an overestimation for an atmosphere- wide value. However, the amount of CO2 being created from the oxidation of CH4 is quite minor and as such it doesn't really matter much in this study – it was more a comment for future work. Regarding the OH: here I guess the Spivakovsky fields distributed within the TransCom CH4 project, reduced by 8%, are meant? If so, a reference to Spivakovsky et al. (1990) should be added.

We agree with the reviewer that the assumption of the globally constant CH4 value is optimistic. However, as also pointed out, its effect is very minor, especially near the surface. Furthermore, we consider such a simple treatment of CH4 reasonable, because we performed the CO2-only inversion and did not used CO data in the inversion.

Here, we apologize for incorrect descriptions of the BVOC data of VISIT caused by misunderstanding among co-authors. We now realized that we did not consider the oxidation of BVOC, but only considered the direct emissions of CO from vegetation. Although we recognize that the oxidation of BVOC is important for simulating atmospheric CO, we think it would not significantly affect our results and conclusions in this study, because we performed the CO2-only inversion and focused on biomass burnings.

In this regard, we modified the last part of Section 2.2.2 as follows.

"In the model, the contribution of oxidation from biogenic volatile organic compounds (BVOCs) to CO is not considered yet, but direct CO emissions from vegetation are given at the earth's surface. Although the oxidations of $CH_4$ and BVOCs are significant sources of atmospheric CO, we treated the former very simply and did not consider the latter. Therefore, we did not input CO observations to the inverse analysis. In the inversion, the biomass burning emissions of CO, which were predominant in Equatorial Asia, were modified along with those of $CO_2$, as described in the next section." **[Lines 167-172]**

For the reference of the OH data, we appreciate the reviewer's suggestion. Accordingly, we added Spivakovsky et al. (2000) as the reference **[Line 166]**. Please note that, rather than Spivakovsky et al. (1990), Spivakovsky et al. (2000) might be more appropriate, because the TransCom project used the latter dataset.

I guess that these signals are really small anyhow, but it would good to quantify this, also for BVOCs: approximately how large are the different components in the simulated (prior) signal of CO2? What is essentially background (from outside of your targeted region), how much is from local anthropogenic emissions, how much from oxidation of other species, how much from the biosphere (net), and how much from biomass burning? I would be really interested to see the contribution of the different signals to the total simulated signal, even just for e.g. one simulated flight.

According to the reviewer's suggestion, we made the same figure of Fig. 4, but for different CO2 components; they are from the fossil fuel emissions, the terrestrial biosphere and ocean fluxes, and from the oxidation of CO (Fig. R1). Note that this CO have all the contributions from the fossil fuel emissions, BVOC, and the fire emissions. From this figure, we can clearly see that contributions of the fluxes other than the fire emissions are marginal, indicating that fire signals are dominant in the observed mole fraction variations over Singapore. Of course, as the model resolution is limited, we cannot exclude possibilities of contributions from local fossil emissions or natural terrestrial biosphere fluxes. Based on this figure, we added a sentence as below.

"Nevertheless, a model simulation that separately calculated $CO_2$ mole fractions from other different sources (fossil fuel emissions, terrestrial biosphere and ocean fluxes, and oxidation of CO) indicated that these fire contributions are dominant in the $CO_2$ mole fraction variations over Singapore for this period." **[Lines 295-297]**

[Figure]

**Fig. R1**. The same as Fig. 4 of the manuscript but for different CO2 components: from the fossil fuel emissions (a), the terrestrial biosphere and ocean fluxes (i.e., GPP+RE+ocn) (b), and from oxidation of CO (c).

I also had a question about Figure 8 and 9 that I would like to see addressed in the text: what do you think is the reason for the large positive adjustment over the Indochinese Peninsula (Vietnam,

Cambodia, Laos. . .) for this period? This is not seen in the fire prior at all, and I presume that it is a biosphere signal, but the adjustment is really large for both months, but in opposite directions. Indeed, it almost looks as if the prior for October better fits the posterior for September, and the prior for September the posterior for October. Is this a shifting in the seasonality of the biospheric fluxes from the climatological norm (or the 2003-2005 mean) due to El Nino? Some discussion of this would be welcome.

We appreciate the reviewer for pointing out the interesting feature in the posterior flux patterns. Indeed, it would be worth investigating that flux pattern change, because the CONTRAIL aircraft flew to Bangkok frequently in 2015 and the observations may have constrained the fluxes. However, in this paper, we focused only on Equatorial Asia, which does not include Indochinese Peninsula, and we also focused on fire emissions. As the fire season of Indochinese Peninsula is completely different from that of Equatorial Asia, those flux changes might be contributed by biospheric fluxes; however, we need a further analysis, which is left for a future study.

Finally, no estimate of the uncertainty of the resultant fluxes is presented, unless one considers the spread related to the different priors. Is such an estimate feasible with the inversions system presented here? Even if such a calculation is not technically feasible, some discussion of this shortcoming should be included. And one last minor point: I think it would be beneficial to explain earlier in the text (in the methods section) how the emission ratio of CO/CO2 is set. References to Akagi et al. only come much later (in the discussion).

In fact, estimating posterior uncertainties is feasible with our inversion system and we have described its algorithm in a recently published paper (Niwa and Fujii, 2020). However, it is a little bit computationally demanding, especially for the resolution we used here (dx~112km). In the last part of the Discussion section, we added some discussion as below.

"In this inversion, we did not calculate the posterior errors, which could give implications of the estimated flux uncertainties. Instead, the spread of the sensitivity tests shows a reasonable range of conceivable flux estimates. In fact, an algorithm for estimating posterior errors was developed by Niwa and Fujii (2020) and it could be applicable to the inverse calculation. However, we left it for a future comprehensive inversion because the algorithm is computationally demanding, especially for the model resolution we used here." **[Lines 568-562]**

For the emission ratio, accordingly, we added a description of the emission factors of CO2 and CO in Section 2.2.5 as follows.

"Note that both the datasets use similar emission factors of $CO_2$ and CO based on Akagi et al (2011); particularly, the same emission factor from Christian et al. (2003) was applied to peatland, from which a large part of fires arises in Equatorial Asia (van der Werf et al., 2017)." **[Line 245-247]**

Besides this, I made some language corrections/suggestions in the document itself, which I have uploaded. I hope that some of these may be helpful.

Please also note the supplement to this comment: https://acp.copernicus.org/preprints/acp-2020-1239/acp-2020-1239-RC2- supplement.pdf

We really appreciate the corrections/suggestions the reviewer made. All of them are incorporated in the revised manuscript.